# The Stability Study of Cefepime Hydrochloride in Various Drug Combinations

**Joanna Żandarek** [1,2]**, Żaneta Binert-Kusztal** [1]**, Małgorzata Starek** [1] and **Monika Dąbrowska** [1,*]

1   Department of Inorganic and Analytical Chemistry, Faculty of Pharmacy, Jagiellonian University Medical College, 9 Medyczna St., 30-688 Kraków, Poland
2   Doctoral School of Medical and Health Sciences, Jagiellonian University Medical College, 16 Łazarza St., 31-530 Kraków, Poland
*   Correspondence: monika.1.dabrowska@uj.edu.pl

**Abstract:** Modern antibiotics face many obstacles, starting with the ever-increasing resistance of microorganisms directed against the antibiotic. An important problem is also the existing trend of polypharmacy. The aim of this study was to develop qualitative and quantitative conditions for the determination of cefepime-hydrochloride solution individually and in mixtures containing other substances with biological activity, such as ketoprofen, gestodene with ethinylestradiol, estradiol, caffeine, calcium ions, paracetamol, bisoprolol, acetylsalicylic acid and ibuprofen, using thin-layer chromatography combined with densitometric analysis. The influence of temperature on the stability of cefepime in these situations was investigated. Furthermore, the effect of UV radiation on the stability of the antibiotic in model drug mixtures was tested. On the basis of the dependence of changes on the concentration of cefepime over time, the order of the reaction was designated, followed by the kinetic parameters of the reactions. Statistical analysis proved that the rate-of-concentration changes in the analyzed conditions corresponded to first-order kinetics. In the course of optimizing the analytical procedure, taking into account the lack of interference of the main peak with the additional peaks and the retardation factor ($R_F$), the mobile phase with the composition of ethanol: 2-propanol: acetone: water (4:4:1:3, $v/v/v/v$) was selected, while silica gel 60F$_{254}$ TLC plates were used as the stationary phase. Cefepime-peak areas obtained during the analysis at the analyzed time points allowed us to conclude that the stability of the antibiotic decreased with increasing temperature. The greatest stability was obtained in mixtures with calcium ions (half-life values ($t_{0.5}$) up to 1320.00 h), while the greatest degradation occurred in combination with hormones ($t_{0.5}$, 2.00 h at 40 °C). Studies have also demonstrated the destructive UV-radiation impact on the stability of these antibiotic-drug combinations ($t_{0.5}$, 0.23–0.71 h).

**Keywords:** cefepime; stability testing; TLC with densitometric detection

## 1. Introduction

Antibiotic resistance has now become a significant public-health problem worldwide. Due to microbial resistance to antibiotics, infections that would normally respond to antibiotic treatment have become difficult or even impossible to treat. This results in increased morbidity, failure in treatment, higher mortality and costs to society [1]. Unfortunately, the resistance of bacteria to different classes of antibiotics is constantly increasing. As a result, the hitherto-effective antibiotics are losing their importance, and the health situation is approaching the pre-antibiotic era. The value of antibiotics as life-saving drugs is under threat [2]. The relationship between antimicrobial resistance and antimicrobials use is well documented. However, little information is available on the antimicrobials used in low-income countries. In most countries, the most commonly used antibiotics are amoxicillin and amoxicillin/clavulanic acid. These substances fall under the availability category of the Model List of Essential Medicines, which includes antibiotics recommended

as first- or second-line drugs for common infections, and which should be available in all countries [3]. Broad-spectrum antibiotics such as third-generation cephalosporins, carbapenems and quinolones are classified as antibiotics that should be used with caution, due to their high resistance potential and/or their side effects [4]. In 2017, 0.6 million new cases of tuberculosis were reported that were resistant the most effective first-line drug, rifampicin, of which 82% were multidrug resistant to *Mycobacterium tuberculosis* and did not respond to isoniazid and rifampicin. Data on antimicrobial consumption in the European region is collected by two separate networks that share common methods to collect and analyze data on antimicrobial consumption. ESAC-Net is a network of national surveillance systems that provide reference data on the use of antimicrobials [5]. Data presented by the WHO (World Health Organization) in 2015 on antibiotic consumption in a group of 45 countries and Kosovo, expressed as defined daily doses per 1,000 inhabitants (DDD), suggest that the most commonly consumed drugs were penicillins (7.1 DDD). Consumption of macrolides/lincosamides/streptogramins was 2.6 DDD per 1000 inhabitants per day, the lowest in Sweden (0.7 DDD) and the highest in Greece (7.7 DDD), accounting for 5% and 23% of the total intake, respectively. Consumption of quinolones and tetracyclines was similar, and amounted to 1.8 DDD and 1.6 DDD per 1000 inhabitants per day, respectively, while for the sulfonamides and trimethoprim subpopulations the average was 0.6 DDD. Monitoring antimicrobial consumption is a key component of the national and local antimicrobial-management program [6]. A summary of data on antibiotic resistance and information on the amount and method of their consumption helps to identify areas for development, targeting, monitoring and evaluation of these activities in the field of the presented problem. Managing antibiotics is an effective way to use medications wisely. The pharmacist plays an important role here, by monitoring medications, consulting with doctors and patients, and contributing to the prudent use of antibiotics [7].

1945 was a turning point in the history of the fight against microorganisms, when Fleming received the Nobel Prize for the discovery of penicillin (in 1928), while in Italy Giuseppe Brotzu discovered the fungus *Cephalosporinum acremonium* with relatively little activity against *Salmonella paratyphi*, *Salmonella typhi*, *Vibrio cholerae*, *Staphylococcus aureus*, *Brucella melitensis* and *Brucella melitensis*. This gave rise to further research, and in 1948 cephalosporin P was separated, followed by cephalosporin N and cephalosporin C. In 1961, the chemical structure of these antibiotics was identified, which contributed to the introduction of cephaloridin as the first cephalosporin (in 1962). Due to its high nephrotoxicity, it is currently not used in medicine [8]. These discoveries led to the creation of a new generation of cephalosporin antibiotics that saved many lives. However, soon after their introduction into clinical practice, almost all of them developed resistance [9]. Cephalosporins, by inhibiting the synthesis of the microbial-cell-wall structure by inhibiting enzymes belonging to penicillin-binding proteins, i.e. carboxypeptidases, endopeptidases and transpeptidases, have a bactericidal effect.

Penicillin-binding proteins (PBP) are responsible for the formation of the main component of the wall, murein, which is formed by N-acetylmuramic acid and N-acetylglucosamine. By attaching to the active site of the enzyme, the antibiotic inhibits peptidoglycan elongation. As a result, the osmotic pressure increases, the activity of intracellular hydrolases increases, the synthesis of the bacterial wall is inhibited, and autolytic processes are intensified, which results in cell death. The most important element from the point of view of antibiotic activity is the β-lactam ring. In relation to gram-negative bacteria, the activity of cephalosporins is hindered by the presence of the periplasmic space and the membrane barrier located on the outer side. Antibiotics in this group are characterized by high tolerance and low toxicity. The most common side effects of these medications include an allergic reaction, with maculopapular rash, fever, bronchospasm, urticaria, and even anaphylactic shock. However, this reaction is less common than with penicillin. Among the penicillin-hypersensitive population, 20% are also allergic to cephalosporins. Another possible effect which increases with low albumin levels, kidney failure, and higher drug intake, is a positive Coombs test, characterized by the formation of red-blood-cells aggregates.

Cefamandol, cefotene, cephalotin and cefoperazone, through the N-methylthiotetrazole substituent present in their molecules, cause ethanol intolerance, changes in coagulation and bleeding time. Side effects are also related to the way the cephalosporin is administered. Intramuscular administration may result in abscesses and severe pain, intravenous injection is associated with a higher risk of thrombotic vasculitis, while administration into the cerebrospinal fluid may lead to disturbances in the proper functioning of the nervous system [8]. Cephaloridine nephrotoxicity has become less important, due to the emergence of new generations of drugs with much less kidney-damage potential. However, there are situations when renal function should be monitored, mainly in patients treated concomitantly with aminoglycosides and cephalosporin antibiotics or taking cephalosporins alone, but at much higher doses. Allergies are observed in 1–4% of patients, while anaphylactic shock is particularly prevalent in patients treated with third-generation antibiotics [10]. Chemically, penicillins have a β-lactam ring attached to a thiazolidine ring with one side chain. The penicillin-mediated IgE-reaction (type one allergic reaction) probably involves all antibiotics containing a β-lactam ring. However, it is suspected that non-IgE-mediated penicillin allergy is often related to the side-chain, and may be the source of delayed rashes, whereas, cephalosporins have a β-lactam ring linked to a dihydrothiazine ring with two side chains. Allergic reactions to cephalosporins are more often attributed to the antigenic activity of the side-chain than to the ring itself. The side-chain of older members of a cephalosporin (e.g., cephalotin, cephaloridin, cefamandol) is penicillin-like, which increases the risk of cross-reaction and type-four allergic reactions to penicillin and its derivatives [11].

Cefepime, a representative of the fourth generation of cephalosporins, exhibits a broader spectrum of activity against gram-positive bacteria. In addition, due to the possibility of taking the zwitterionic form, cefepime easily penetrates through the pores of gram-negative bacteria located on the outer side of the cell membrane, which extends the therapeutic effectiveness of antibiotics of this class, compared to the previous generation, acting on the lower respiratory tract, urinary system, lungs, prostate, soft tissues, abdominal cavity and gallbladder. It is resistant to numerous chromosomal and plasmid β-lactamases. As an enzyme inducer, it has a relatively weak effect on one type of β-lactamases. The antibiotic can be administered both intravenously and intramuscularly, where the absorption of the drug is rapid, with about 100% bioavailability. It is excreted from the body unchanged, almost exclusively by the kidneys [8].

The ability to treat fever and inflammation dates back to 3500 BC, when the Greek physician Hippocrates used willow-bark extract. It was not until the 17th century that the active ingredients of the bark were identified in Europe as salicylates [12]. Currently, non-steroidal anti-inflammatory drugs (NSAIDs) are a family of commonly consumed drugs with potent anti-inflammatory, analgesic and antipyretic properties. They have also been shown to prevent or treat heart disease, certain cancers, and Alzheimer's disease, in part because of the common inflammation associated with these pathogenic conditions. One of the main concerns with the chronic use of these drugs is their toxicity, especially in the case of gastrointestinal ulcers. The molecular basis of both the benefits and pathogenesis of these drugs is attributed to their ability to inhibit the pro-inflammatory enzyme cyclooxygenase. Prescribing NSAIDs to the elderly should be based on the lowest effective dose for the shortest duration. In this case, comorbidities and decreased liver function may affect their pharmacokinetics [13,14]. Paracetamol (the name used in Europe) or acetaminophen (the name used in the United States) is the most widely used antipyretic agent in the world. The offer includes about 100 products, which contain paracetamol alone or in combination with other active ingredients. In the WHO analgesic ladder, this drug was placed at all three levels of pain-intensity management. However, when taken in doses exceeding the therapeutic dose, it poses a high risk of liver damage [15]. In various types of pain of moderate intensity, paracetamol as a weak analgesic, along with NSAIDs or coanalgesics (e.g., caffeine) has been used as the basic non-opioid analgesic. Acetaminophen is the recommended first-choice oral analgesic for long-term use and the drug of choice in patients for whom NSAID are contraindicated. About 90% of this drug is metabolized

in the liver to inactive the conjugates of glucuronic acid, sulfuric acid and cysteine, and is excreted in the urine. The rest is N-hydroxylated in the liver to form the toxic metabolite of the imine N-acetylbenzoquinone, which is very quickly inactivated by the glutathione sulfhydryl groups, and excreted as mercapturic acid. Paracetamol poisoning is treated with N-acetylcysteine [16,17].

Bisoprolol is a β1-adrenoceptor antagonist that has been shown to lack a partial membrane agonist or stabilizing effect. They showed that it is a more potent β1-adrenergic antagonist than either atenolol or metoprolol, but it appears to be less potent than propranolol and betaxolol, in this respect. Bisoprolol has a long duration of action, with a marked reduction in exercise tachycardia (approx. 20%) in people with stable angina. Bisoprolol, which has both lipophilic and hydrophilic properties, has been shown to have an oral bioavailability of >90%, is rapidly and widely dispersed, but only slightly crosses the blood–brain barrier. About 50% of the dose is metabolized in the liver to three inactive metabolites, while the rest is excreted unchanged by the kidneys [18].

Sex hormones play an important role in regulation of physiological processes, including blood pressure. Therefore, they may be important factors in the development of hypertension and coronary-artery disease. For example, high endogenous estrogen levels in premenopausal women are associated with a lower risk of many diseases such as hypertension, diabetes, obesity, vascular disease, and stroke, compared to men of similar age [19]. Hormonal contraceptives are used in clinical practice to prevent ovulation and implantation and the possible penetration of sperm into the egg. They are most commonly available as oral formulations containing a fixed combination of a synthetic estrogen (ethinylestradiol) and a progestogen (gestodene) [20]. Combined progestin–estrogen contraceptive pills are used by women around the world, and provide a reliable, reversible method of contraception. Gestodene has a positive pharmacokinetic profile, characterized by high bioavailability and a prolonged half-life, which contributes to contraceptive effectiveness. It is the most potent of all orally administered progesterone derivatives in inhibiting ovulation. In addition, it has minimal effects on lipid and carbohydrate metabolism, and its hemostatic effect appears to reflect the balance between clotting, and fibrinolytic factors [21,22].

Calcium is the most abundant mineral in the human body. It is important for intracellular metabolism, bone growth, blood clotting, nerve and muscle conduction, and heart function. A total of 99% of the calcium in the body is bound to the bones, and only 1% is freely exchanged with the extracellular fluid. To maintain calcium at a constant level, the cooperation of three hormones is necessary: parathyroid hormone, vitamin D and calcitonin [23]. In daily practice, healthcare professionals are faced with an increasing number of patients taking calcium supplements without proven deficiencies, especially postmenopausal women. This practice has spread rapidly over the past decade, based on the purported role of calcium in preventing osteoporosis and fractures, especially hip fractures, among the elderly population [24].

Caffeine acts as a non-selective competitive phosphodiesterase inhibitor. It blocks adenosine receptors, mainly A1 and A2, and causes an increased release of dopamine, norepinephrine and glutamine. It can induce the release of calcium from the sarcoplasmic reticulum and inhibit its reuptake. Some of the caffeine-induced effects may be mediated in part by modulation of neuromuscular function and an increase in the strength of skeletal-muscle contraction. A potential reverse effect of caffeine is to stimulate diuresis. Caffeine is now believed to be the most widely consumed psychostimulant in the world. It is found in many prescription and over-the-counter (OTC) medications, including headaches, colds, appetite control, stimulants, asthma and edema. People take very large doses of caffeine, ignoring any safety concerns, by taking combinations of two or more sources of caffeine, such as medications and energy drinks. The main symptoms of excessive caffeine consumption are restlessness, agitation, insomnia, gastrointestinal disturbances, tremors, tachycardia, psychomotor agitation, and in some cases even death [25].



The use of reliable and effective drugs is becoming more and more complicated. With the increase in the number of active substances used by humans, the possibility of drug–drug interactions, which may have clinically significant consequences, also increases [26].

A drug–drug interaction can cause an unexpected side effect. For example, mixing a sedative drug and allergies, e.g. antihistamines, can slow down reactions. Some drug interactions can even be harmful. Simultaneous intake of many drugs can significantly change the kinetics and pharmacodynamics of individual substances, intensifying or weakening their action, which contributes to the failure of the therapy, and the occurrence of numerous side effects. Furthermore, proper storage of medicaments also affects their durability and effectiveness [27]. Standard stability studies do not contain stability testing of medications in combination with others, except in the case of formulations containing several active substances. Therefore, addressing this issue seems to be an important aspect of further research.

Scientists are dealing with the problem of drug degradation under the influence of other therapeutic compounds present in pharmaceutical products. For example, Maswadeh describes a study of the total effect of paracetamol with cefuroxime-axetil suspension as the combination of choice for children and adolescents. He used the infrared-spectroscopy method, which showed the effect of both paracetamol and auxiliary substances on the absorption spectrum of the cephalosporin. Finally, it was concluded that oral administration of these drugs in combination may affect the physicochemical properties, dissolution rate, solubility, and the bioavailability of one or both drugs [28]. Other researchers focused on elucidating differences in the interaction of chiral ibuprofen and naproxen with the drug excipient (poly(vinylpyrrolidone)), in the solid state. Physical-drug mixtures (poly(vinylpyrrolidone) and solid dispersions were analyzed by scanning electron microscopy, Fourier-transform infrared spectrometry, solid-state $^{13}$C NMR spectroscopy and X-ray diffraction. Significant differences in the physicochemical properties of the analyzed model preparations were found. The authors detected spontaneous conversion of the ibuprofen-poly(vinylpyrrolidone) physical mixtures to a stable glassy form during storage. The enantiomer reacted more strongly than the racemate, while naproxen did not interact with the drug excipient [29].

There are many factors that can degrade or alter the active ingredients in pharmaceutical formulations when used together. The research carried out as part of the presented work focused on four points: (i) preparation of multicomponent mixtures of cefepime with other biologically-active substances included in the research plan, (ii) development and optimization of the conditions for the separation and identification of cefepime in multicomponent model mixtures by thin-layer chromatography combined with densitometric detection, (iii) quantitative analysis of cefepime in various mixtures of model drugs at different time points, under changing temperature conditions, and (iv) determination of the influence of UV radiation on the stability of the analyzed antibiotic.

The developed methodology may be useful for further research on the expected drug interactions between the cephalosporin antibiotic cefepime hydrochloride and other drugs commonly used by people treated, e.g. for pain and chronic diseases. This is very important from the point of view of possible interactions of concomitantly used active substances that may affect the required therapeutic dose of the drug, ultimately weakening or enhancing its effect.

## 2. Materials and Methods

### 2.1. Chemicals and Apparatus

Standard substances: cefepime HCl (CDS022365-100MG), Sigma Aldrich, Poznań, Poland), ketoprofen (K1751), ibuprofen (I7905), and caffeine (C0750) were purchased from Sigma Aldrich, Poznań, Poland.

Preparations: Estrofem Serie No. GF70075, (Novo Nordisk, Zürich, Switzerland), Femoden No. 51005A (Bayer Pharma AG, Leverkusen, Germany), Aspirin (Bayer, Leverkusen, Germany), Bibloc Serie No. ES7489 (Sandoz sp. z o.o., Warszawa, Poland),

Paracetamol (USP Zdrowie Sp. z o.o., Warszawa, Poland), and Calcium gluconicum Serie No. 1710024 (Farmapol, Poznań, Poland) were obtained in the local pharmacy

TLC Silica gel plates 60 $F_{254}$, No. 1.05554.0001 (Merck, Darmstadt, Germany) were used as a stationary phase. All reagents: acetone (Stanlab, Lublin, Poland), ethanol anhydrous 99.8% (POCh, Gliwice, Poland), 2-propanol (Merck, Darmstadt, Germany), and methanol (Chempur, Piekary Śląskie, Poland) were of analytical grade.

Densitometer TLC Scanner 3 with Cat4 software (Muttenz, Camag, Switzerland), Linomat V (Muttenz, Camag, Switzerland), analytical balance WPA 120C1 (Radom, Radwag, Poland), micro-syringe (Hamilton, OH, USA), and dryer EcoCell BMT (Brno, Czech Republic) were used.

### 2.2. Sample Preparations

Stock solutions of all the studied compounds were prepared in methanol at concentration 10 mg/mL. In the case of standard substances, equivalents of 50 mg of free base were weighed into the 5.0 mL volumetric flasks, which were then filled with methanol. In the case of pharmaceutical formulation, firstly the tablets were ground in a mortar, then equivalents of 50 mg of each substance were weighed into the 5.0 mL volumetric flasks, which were then filled with methanol. After 5 min of ultrasonic sweeping, the suspensions were centrifuged at 15,000 rpm for 5 min. Obtained supernatants were used as the stock solutions. Working solutions, containing 100 µg/mL of each substance, were prepared by diluting the stock solutions with the gradient-grade water. The obtained working solutions were submitted to the irradiation procedure.

### 2.3. Optimization of Determination Conditions

The suitability of the fully validated method previously developed by our team to determine the content of cefepime hydrochloride was checked [30]. In order to determine the optimal conditions for the separation of cefepime in model mixtures and in the presence of possible degradation products, mobile phases consisting of ethanol: 2-propanol: acetone: water in various volume ratios: 4:4:1:3 and 4:4:3:1 and 4:4:2:2 (*v/v/v/v*), were tested.

Based on the obtained data (retardation factors, $R_F$), a mobile phase was selected with the following composition: ethanol: 2-propanol: acetone: water (4:4:1:3, *v/v/v/v*), ensuring the separation of individual components of the analyzed mixture and the lack of interference of the main peak with possible degradation products formed during the tests, which allowed for the quantification of the antibiotic in tested combinations.

### 2.4. Preparation of Model-Drug Mixtures

In order to study the behavior of cefepime (CEF) in mixtures, possible combinations of drugs consumed by patients (such as ketonal (KET), calcium (CALC), gestodene and ethinylestradiol (GES + ETHYN), estradiol (EST), bisoprolol (BIS), caffeine (KOF), paracetamol (PAR), acetylsalicylic acid (ASA), and ibuprofen (IBU)) with concomitant cephalosporin therapy, were analyzed. Then, 16 drug mixtures ('CEF, KET', 'CEF, KET, CALC', 'CEF, 'CEF, EST', 'CEF, EST, KET', 'CEF, GES + ETHYN, KET', 'CEF, GES + ETYN, CALC', 'CEF, BIS', 'CEF, KOF', 'CEF, KOF, PAR', 'CEF, KOF, ASA', 'CEF, ASA', 'CEF, BIS, PAR', 'CEF, BIS, IBU', 'CEF, BIS, CALC', 'CEF, EST, CALC') and a comparative mixture containing cefepime were prepared. For this purpose, each solution of the mixture component was pipetted in a volume of 0.5 mL into glass ampoules, which were then sealed (so as to test the effect of temperature on the stability of cefepime). Mixtures in quartz vessels were prepared analogously for the analysis of the effect of UV radiation (254 nm).

### 2.5. Chromatographic Conditions

The influence of different temperatures (4, 40, 60, 80 °C) at different time points (0–2 h for higher temperatures and 1.5 month for 4 °C) on the stability of the cefepime was tested. The influence of ultraviolet radiation on the stability of the antibiotic was also investigated. Simultaneously with the irradiation, a comparative analysis of the control samples was

carried out under a UV lamp without radiation. Then, at each time point, 5 μL of each mixture in question was loaded onto TLC plates of silica gel $60F_{254}$ with the Linomat V. The plates were dried at room temperature, and then developed to 95 mm in mobile phase, in a chromatographic chamber of size $18 \times 16 \times 8$ cm (Sigma-Aldrich, Laramie, WY, USA). The chamber was first saturated (15 min) at room temperature, with the selected mobile phase. After the plates were developed, they were air-dried at room temperature. The obtained chromatograms were subjected to further analysis. The obtained results were assessed visually (by visualizing the spots under a UV–Vis lamp, at 254 and 366 nm) and densitometrically (scanning speed 20 mm/s, slit dimensions $4.00 \times 0.45$ mm, over the range of 200–400 nm) recording the absorption spectra.

*2.6. Statistical Analyses*

The analysis was carried out using Statistica v 13.3. TIBCO Software Inc. The confidence limit of $p < 0.05$ was considered as statistically significant.

### 3. Results

An analysis of the available literature provided information on the possible influence of different factors on the stability of cephalosporin antibiotics. Therefore, we decided to investigate the stability of cefepime in combination with various active substances that could potentially be consumed during cefepime therapy, such as hormones regularly taken by women as part of hormone-replacement therapy (HRT), gestodene and estradiol used in contraception, and bisoprolol as a cardioselective β-adrenergic receptor antagonist in heart disease, under various environmental conditions. The impact of ad hoc medications, with a wide range and easily available to the population, such as NSAIDs with anti-inflammatory and analgesic properties (e.g., ketoprofen, acetylsalicylic acid, ibuprofen) was also analyzed. Other preparations under consideration include paracetamol, which is not considered an NSAID due to the lack of an anti-inflammatory component, and calcium or caffeine, which enhance the activity of acetylsalicylic acid, and paracetamol, often used by the pharmaceutical industry to create complex formulations.

Mixtures of the tested active substances were subjected to chromatographic analysis as part of the previously developed and validated thin-layer-chromatography procedure with densitometric detection [30]. On the basis of the recorded surface-area values for cefepime, the corresponding concentration expressed as a percentage was calculated, and the natural logarithm was determined. Next, using Statistica software, the order of the antibiotic-degradation reactions, in particular conditions, in given drug mixtures, was calculated. Under the conditions described earlier, the correct separation of the components of the analyzed mixtures was found. Furthermore, under the described conditions the required sensitivity and symmetrical peaks were obtained, which allowed the use of this procedure for the analysis of substances included in the test plan.

The concentration of cefepime was expressed using the chromatographic-densitometric technique, by the following formula:

$$[\%]x = \frac{Pp. \times 100}{Pw.}$$

where: $x$—concentration [%], $Pp.$—peak area registered after incubation [mm$^2$], $Pw.$—peak area registered before incubation [mm$^2$].

Each measurement was performed three times, and the average value was taken as the result. TLC is not complicated, and is a fast analytical procedure which can be successfully used both for the separation and qualitative and quantitative analysis of various substances, and allows the carrying out of several analyses simultaneously. Due to the simplicity of conducting the analysis, the low costs, the need to minimally purify the sample before analysis, and additionally, the possibility of multiple samples being analyzed at the same time, it allows various types of research to be conducted, and thus it is used in miscellaneous areas of science.

The cefepime hydrochloride solution at 4 °C is relatively stable over an incubation period of 3 weeks, during which the concentration of the antibiotic drops, by about 25% from the initial amount. Further observation showed a decrease in the concentration of the antibiotic (after a 1.5-month incubation) by 60%. In the compositions with hormones, the lowest degradation was found in the model mixture with the composition of 'CEF, EST, CALC', where the amount of cefepime after 55 days was 43%, while the least-stabilizing combinations were 'CEF, EST, KET' and 'CEF, GES + ETHYN, KET', in which the concentration of the antibiotic after the same observation period fluctuated around the minimal measurable level of 4–5%. Under analogous time and temperature conditions, the mixture of 'CEF, KET, CALC' and 'CEF, BIS, CALC' showed a large, stabilizing effect on cephalosporin; its concentration on the 22nd day of the study was in the range of 92–98%. Further incubation showed that the least-stable mixture was 'CEF, EST, KET', the content of which dropped to 18% after another week of incubation. After the entire period of observation of the mixtures at 4 °C, only the system containing 'CEF, BIS, CALC' remained at the level of 50% from the baseline value, and after 3 weeks of observation the amount of antibiotic in this system decreased by only 10%. The addition of OTC drugs does not increase the durability of the antibiotic. The most durable combination in this case is 'CEF, KOF, PAR', in which the amount of cefepime decreases by 30% after 22 days of analysis, while the combination with caffeine and acetylsalicylic acid proved to be the least-stable antibiotic. In this case, after 14 days of analysis, 45% remained, and after 55 days, only 7% of the content on the day of the mixture preparation (Table 1). The standard deviation (Sd) for all changes in the concentration (%) of cefepime ranged from 0.32 to 1.52, which is graphically presented in Figure 1.

**Table 1.** Concentration [%] changes of cefepime in various drug mixtures during incubation at 4 °C ($n = 3$).

| Mixture | 336 [h] | | 528 [h] | | 1320 [h] | |
|---|---|---|---|---|---|---|
| | Xm | Sd | Xm | Sd | Xm | Sd |
| CEF | 75.01 | 0.32 | 72.09 | 0.69 | 40.23 | 1.06 |
| CEF, KET | 58.32 | 0.46 | 46.84 | 0.84 | 17.48 | 0.75 |
| CEF, KET, CALC | 92.45 | 0.28 | 67.14 | 0.68 | 36.45 | 1.18 |
| CEF, GES + ETHYN | 40.21 | 0.43 | 30.78 | 0.70 | 9.22 | 1.24 |
| CEF, ESTRAD | 54.01 | 0.52 | 35.34 | 0.49 | 12.36 | 1.00 |
| CEF, EST, KET | 36.45 | 0.30 | 18.21 | 0.37 | 4.51 | 0.59 |
| CEF, GES + ETHYN, KET | 36.67 | 1.00 | 20.07 | 0.63 | 5.11 | 0.90 |
| CEF, GES + ETHYN, CALC | 89.54 | 0.76 | 57.91 | 0.66 | 29.42 | 1.20 |
| CEF, BIS | 60.68 | 0.31 | 53.20 | 0.63 | 21.33 | 1.67 |
| CEF, COF | 70.62 | 1.06 | 66.11 | 0.93 | 29.81 | 1.61 |
| CEF, KOF, PAR | 72.23 | 0.56 | 71.72 | 0.900 | 36.14 | 1.10 |
| CEF, COF, ASA | 45.43 | 0.48 | 38.82 | 0.57 | 7.36 | 0.87 |
| CEF, ASA | 51.23 | 0.82 | 44.41 | 1.33 | 11.62 | 1.36 |
| CEF, BIS, PAR | 64.56 | 0.49 | 60.88 | 1.52 | 33.05 | 1.35 |
| CEF, BIS, IBU | 40.32 | 0.57 | 29.14 | 0.72 | 7.34 | 0.95 |
| CEF, BIS, CALC | 98.12 | 0.48 | 89.66 | 0.92 | 50.83 | 1.12 |
| CEF, EST, CALC | 100.00 | 0.35 | 84.41 | 1.35 | 43.31 | 1.86 |

Where Xm—arithmetic mean, Sd—standard deviation.

The highest stability of cefepime among the analyzed conditions was observed at the temperature of 40 °C, where, after a 2-h incubation, the concentration of the antibiotic remained at an average of 50% from the baseline value. The concentration of the model cefepime solution after 1.5 h analysis under the same conditions decreased only by 20% from the initial value. The surface areas of the antibiotic in mixtures with ketoprofen, regardless of coexisting other substances, dropped to the range of 55–70%. The influence of the analyzed temperatures on the durability of the combination of cefepime with hormones (gestodene and estradiol) was observed (a decrease in the content at the level of half of the

initial value). The addition of caffeine had a stabilizing effect on the antibiotic (83.68%) after 2 h of incubation. A similar relationship was recorded for paracetamol and other model mixtures containing acetaminophen as an additional component. In combination with NSAIDs, the concentration of cefepime dropped by 25%, which was also observed in the presence of bisoprolol, paracetamol or caffeine. Adding calcium ions to mixtures containing as an additional component estradiol or bisoprolol increased the durability of both by about 25% (Table 2).

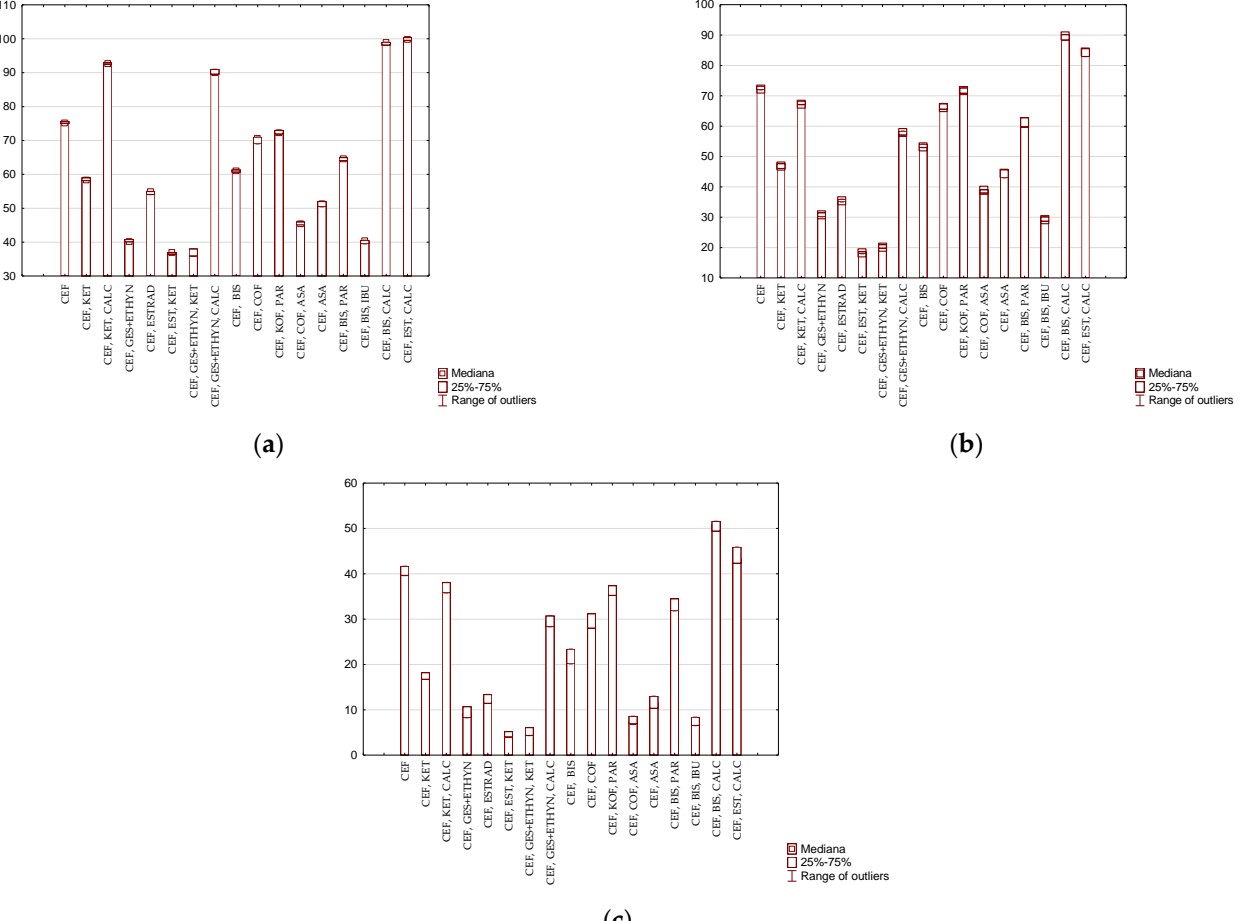

**Figure 1.** An example of graphical standard-deviation plots for experimental data (concentration changes in %) is presented in Table 1. (**a**) Sd for results obtained after 336 h, (**b**) 528 h, and (**c**) 1320 h.

An increase in temperature by another 20 °C has a destabilizing effect on the antibiotic. This is well illustrated by the example of the cefepime standard solution, the content of which, in relation to the initial one, after 2 h at 40 °C decreased by 40%, and at 60 °C to 30%. The highest stability of the antibiotic was demonstrated in the systems of 'CEF, BIS, CALC' and 'CEF, EST, CALC', the concentration of which was in the range of 70–100% throughout the analysis. This may indicate no destructive effect of temperature on the stability of cefepime in both mixtures. As was assumed on the basis of previous stability studies at 40 °C, the most destructive effect on cefepime was shown by gestodene and estradiol and their combinations with ketoprofen, where, in both cases, the degradation kinetics were similar at each measurement point. In both cases the amount of the antibiotic decreased by 87% after 2 h.

**Table 2.** Concentration [%] changes of cefepime in various drug mixtures during incubation at 40, 60 and 80 °C (*n* = 3).

| Mixture | | 0.5 [h] | | | 1 [h] | | | 1.5 [h] | | | 2 [h] | | |
|---|---|---|---|---|---|---|---|---|---|---|---|---|---|
| | | 40 °C | 60 °C | 80 °C | 40 °C | 60 °C | 80 °C | 40 °C | 60 °C | 80 °C | 40 °C | 60 °C | 80 °C |
| CEF | Xm | 98.00 | 90.22 | 71.06 | 88.69 | 80.06 | 66.33 | 80.00 | 77.33 | 43.32 | 62.30 | 30.65 | 18.89 |
| | Sd | 0.82 | 0.39 | 0.85 | 0.36 | 0.84 | 0.62 | 0.95 | 0.55 | 0.64 | 0.59 | 0.85 | 1.22 |
| CEF, KET | Xm | 81.23 | 80.51 | 39.21 | 72.45 | 73.61 | 35.61 | 62.65 | 44.81 | 19.47 | 56.14 | 23.36 | 14.29 |
| | Sd | 0.89 | 0.56 | 0.32 | 0.91 | 0.31 | 0.66 | 0.39 | 0.69 | 1.08 | 0.59 | 0.98 | 1.11 |
| CEF, KET, CALC | Xm | 96.54 | 83.32 | 70.92 | 85.62 | 72.58 | 47.52 | 72.12 | 52.25 | 14.36 | 69.61 | 48.41 | 11.43 |
| | Sd | 0.87 | 0.29 | 0.54 | 0.84 | 0.27 | 0.84 | 0.96 | 0.93 | 1.15 | 0.69 | 0.99 | 0.99 |
| CEF, GES + ETHYN | Xm | 79.62 | 72.59 | 39.76 | 66.93 | 64.36 | 24.36 | 64.96 | 51.36 | 5.40 | 53.45 | 25.26 | 3.08 |
| | Sd | 0.96 | 0.68 | 0.69 | 0.88 | 0.18 | 1.12 | 0.56 | 0.89 | 1.65 | 0.99 | 1.01 | 1.05 |
| CEF, EST | Xm | 76.11 | 71.25 | 52.54 | 74.52 | 63.45 | 36.99 | 69.32 | 38.52 | 5.32 | 51.32 | 20.42 | 4.45 |
| | Sd | 0.59 | 0.91 | 1.09 | 0.69 | 0.76 | 0.99 | 0.81 | 0.79 | 1.58 | 0.86 | 1.23 | 1.08 |
| CEF, EST, KET | Xm | 83.61 | 63.51 | 17.63 | 70.36 | 55.23 | 11.82 | 65.11 | 25.36 | 2.26 | 62.49 | 13.36 | 2.30 |
| | Sd | 0.67 | 0.39 | 1.08 | 0.49 | 1.12 | 1.12 | 0.86 | 1.20 | 1.59 | 0.89 | 1.09 | 1.36 |
| CEF, GES + ETHYN, KET | Xm | 70.21 | 61.82 | 24.58 | 68.33 | 57.63 | 8.96 | 67.36 | 30.25 | 3.15 | 56.26 | 13.52 | 1.02 |
| | Sd | 0.74 | 0.78 | 1.30 | 0.96 | 1.01 | 1.31 | 0.97 | 1.04 | 1.09 | 0.69 | 1.08 | 0.98 |
| CEF, GES + ETHYN, CALC | Xm | 96.09 | 90.85 | 67.63 | 70.68 | 67.58 | 48.62 | 69.84 | 60.31 | 30.09 | 66.52 | 56.79 | 23.13 |
| | Sd | 0.58 | 0.39 | 0.98 | 0.62 | 0.61 | 0.59 | 1.02 | 0.96 | 1.36 | 0.38 | 0.87 | 0.76 |
| CEF, BIS | Xm | 90.58 | 77.62 | 58.84 | 54.63 | 48.74 | 24.89 | 50.63 | 42.42 | 16.15 | 49.73 | 21.36 | 10.04 |
| | Sd | 0.69 | 0.91 | 0.68 | 0.87 | 0.89 | 0.82 | 1.01 | 0.56 | 1.61 | 0.86 | 1.09 | 1.08 |
| CEF, COF | Xm | 92.22 | 78.51 | 62.39 | 86.25 | 75.25 | 57.59 | 84.11 | 72.36 | 40.31 | 83.68 | 60.09 | 30.13 |
| | Sd | 0.36 | 0.89 | 0.67 | 0.95 | 0.98 | 0.39 | 0.96 | 0.98 | 0.96 | 0.59 | 0.84 | 1.01 |
| CEF, KOF, PAR | Xm | 92.31 | 78.88 | 68.12 | 87.69 | 73.92 | 56.47 | 85.82 | 68.26 | 40.09 | 83.98 | 53.88 | 31.54 |
| | Sd | 0.58 | 1.01 | 0.86 | 0.87 | 0.69 | 0.58 | 0.86 | 0.54 | 0.86 | 0.92 | 0.69 | 0.89 |
| CEF, COF, ASA | Xm | 89.11 | 62.36 | 52.36 | 82.24 | 52.36 | 35.02 | 80.09 | 49.63 | 17.61 | 77.56 | 36.26 | 10.26 |
| | Sd | 0.24 | 1.02 | 0.54 | 0.59 | 0.85 | 0.87 | 0.78 | 0.69 | 0.89 | 0.72 | 1.02 | 0.99 |
| CEF, ASA | Xm | 84.25 | 67.85 | 59.45 | 79.52 | 63.85 | 47.56 | 78.36 | 61.42 | 32.09 | 75.51 | 49.89 | 18.42 |
| | Sd | 0.39 | 0.69 | 0.96 | 0.43 | 0.97 | 0.89 | 0.98 | 0.85 | 0.98 | 0.79 | 0.81 | 0.69 |
| CEF, BIS, PAR | Xm | 92.80 | 76.47 | 69.63 | 85.74 | 72.14 | 60.35 | 84.45 | 70.25 | 46.51 | 78.39 | 64.04 | 46.87 |
| | Sd | 0.88 | 0.79 | 0.87 | 0.59 | 0.89 | 0.36 | 0.90 | 0.91 | 0.79 | 0.69 | 0.99 | 0.87 |
| CEF, BIS, IBU | Xm | 92.31 | 65.31 | 40.48 | 80.36 | 56.36 | 23.86 | 79.39 | 52.36 | 10.34 | 76.22 | 40.31 | 9.09 |
| | Sd | 0.92 | 1.01 | 0.39 | 0.63 | 0.79 | 1.12 | 0.99 | 0.84 | 1.09 | 0.82 | 0.69 | 1.12 |
| CEF, BIS, CALC | Xm | 87.12 | 85.56 | 84.69 | 83.29 | 79.69 | 77.36 | 80.24 | 74.45 | 68.51 | 77.36 | 70.52 | 56.21 |
| | Sd | 0.63 | 0.89 | 0.79 | 0.52 | 0.88 | 0.63 | 0.86 | 0.67 | 0.96 | 0.86 | 0.78 | 0.69 |
| CEF, EST, CALC | Xm | 87.36 | 84.25 | 74.88 | 80.36 | 79.57 | 62.43 | 77.36 | 74.11 | 52.39 | 76.55 | 70.36 | 42.25 |
| | Sd | 0.51 | 0.78 | 0.58 | 0.39 | 0.95 | 0.39 | 0.89 | 0.75 | 0.86 | 0.39 | 0.69 | 0.85 |

Where Xm—arithmetic mean, Sd—standard deviation.

Among all the examined temperature conditions, the most drastic for cefepime was its exposure to 80 °C. After one hour, the content of the antibiotic in combination with hormones and ketoprofen decreased by more than 60%, and the subsequent measurement points did not show the presence of the antibiotic to an extent of any therapeutic value.

The peak area of the cefepime in the mixture containing additional bisoprolol and ibuprofen showed a 91% decrease in its content. The addition of caffeine had a variable effect on the content of the cefepime, depending on the composition of the mixtures. In combination with paracetamol, the concentration of cefepime decreased to 30%, while with acetylsalicylic acid it decreased by 90% after 2 h of incubation. The analysis of the mixture containing the β-blocker showed a decrease in the stability of cefepime with increasing temperature. After the 2 h test time, 49% of the antibiotic remained at 40 °C, 21% at 60 °C, and only 10% at 80 °C (Figure 2). The discussed system was more stable at lower temperatures (21% remained at 4 °C after 55 days of analysis). Among all 16 analyzed

model mixtures, the most stabilizing combinations at 80 °C were 'CEF, BIS, PAR' and 'CEF, BIS, CALC', as well as 'CEF, EST, CALC', in which the amount of cefepime in the entire time interval did not drop below 42%. In the remaining analyzed model mixtures, the content of the antibiotic decreased by over 70%.

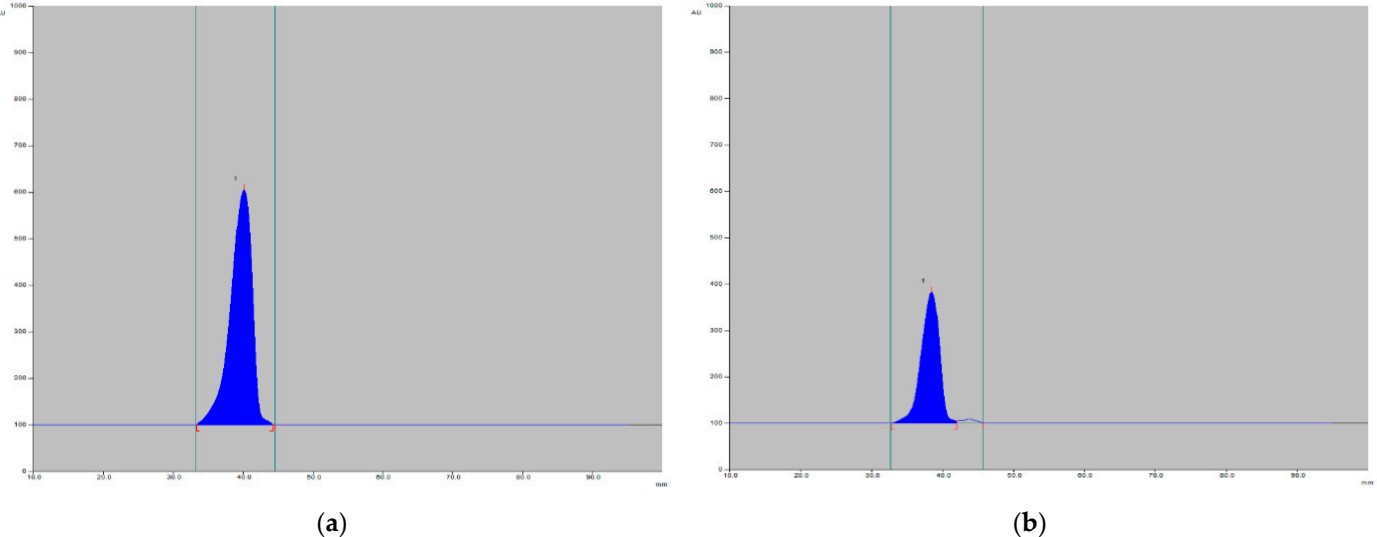

**Figure 2.** An example of a densitogram for 'CEF, GES + ETYN' registered at 80 °C, after (**a**) 0 h, (**b**) 0.5 h.

Bradley et al., in their article on the co-administration of intravenous calcium solutions with ceftriaxone among neonatal patients, noted an interaction in the formation of life-threatening calcium-salt precipitates [31]. The experiments conducted under similar conditions did not confirm this relationship for cefepime. Only after prolonged storage (2 weeks at 40 °C) in mixtures containing calcium ions, precipitation of a white, coarse sediment was observed, probably from the formed calcium salt. Calcium contained in the model mixtures significantly stabilized the cefepime, not leading to sudden decreases in therapeutic concentrations of the antibiotic under all analyzed conditions (Figure 3).

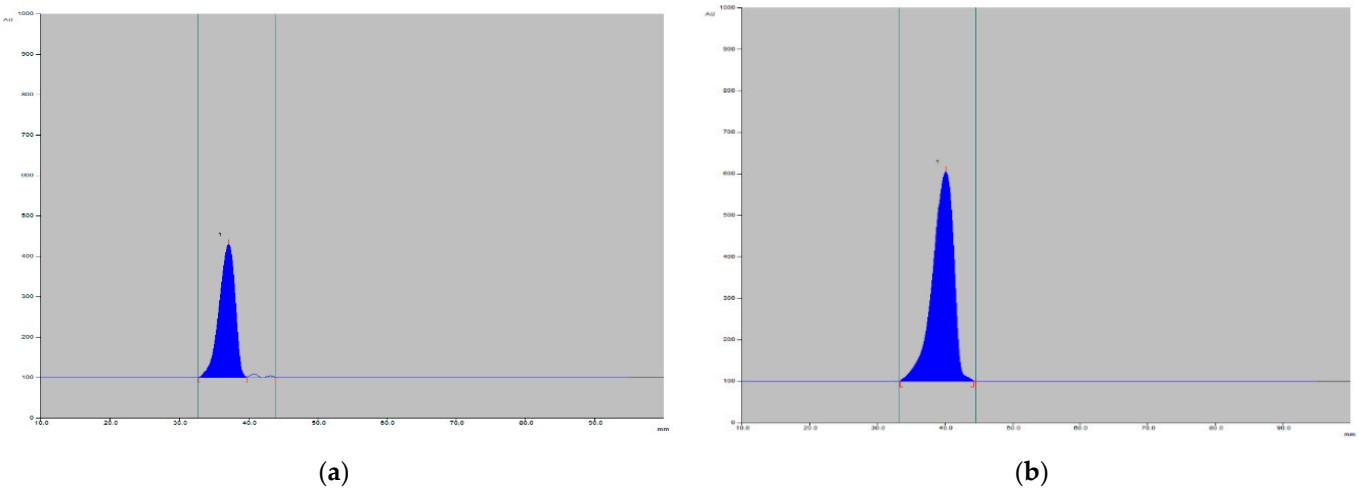

**Figure 3.** Example densitograms recorded for cefepime in a model mixture containing calcium and bisoprolol at a temperature of 80 °C, after (**a**) 0 h, (**b**) 2 h.

The next stage of the work was to test the influence of UV radiation on the cefepime stability in the analyzed solutions. Exposure of cefepime hydrochloride to UV radiation showed its destructive effect on the antibiotic, which was completely degraded after a 2 h

incubation. The fastest decomposition was recorded for the model mixture containing 'CEF, EST, KET', where after 30 min the concentration of the cephalosporin antibiotic dropped to 6% (Table 3).

**Table 3.** Concentration [%] changes in cefepime in various drug mixtures after UV radiation ($n = 3$).

| Mixture | 0.5 [h] | | 1 [h] | | 1.5 [h] | | 2 [h] | |
|---|---|---|---|---|---|---|---|---|
| | Xm | Sd | Xm | Sd | Xm | Sd | Xm | Sd |
| CEF | 78.31 | 0.42 | 39.31 | 0.89 | 23.01 | 0.76 | 8.51 | 1.12 |
| CEF, KET | 28.21 | 0.53 | 13.24 | 0.64 | 9.21 | 1.25 | 4.23 | 1.18 |
| CEF, KET, CALC | 50.36 | 0.81 | 36.54 | 0.88 | 12.32 | 1.10 | 5.36 | 1.53 |
| CEF, GES + ETHYN | 33.51 | 0.39 | 26.77 | 0.79 | 2.25 | 1.44 | 0.00 | – |
| CEF, ESTRAD | 16.45 | 0.82 | 14.63 | 0.58 | 5.09 | 1.20 | 0.00 | – |
| CEF, EST, KET | 6.12 | 1.31 | 5.21 | 1.17 | 0.00 | – | 0.00 | – |
| CEF, GES + ETHYN, KET | 27.32 | 1.20 | 19.36 | 0.83 | 3.25 | 1.50 | 0.00 | – |
| CEF, GES + ETHYN, CALC | 44.89 | 0.96 | 26.24 | 0.66 | 4.31 | 1.32 | 0.00 | – |
| CEF, BIS | 33.54 | 0.89 | 15.51 | 0.83 | 11.21 | 1.07 | 5.31 | 1.39 |
| CEF, COF | 43.64 | 1.12 | 15.11 | 0.68 | 13.98 | 1.41 | 3.81 | 1.61 |
| CEF, KOF, PAR | 59.62 | 0.69 | 37.24 | 0.89 | 16.85 | 1.09 | 14.63 | 0.98 |
| CEF, COF, ASA | 58.01 | 0.57 | 18.36 | 0.71 | 8.03 | 1.27 | 5.54 | 1.03 |
| CEF, ASA | 28.32 | 0.78 | 5.42 | 1.08 | 3.25 | 1.06 | 0.00 | – |
| CEF, BIS, PAR | 43.45 | 0.56 | 23.11 | 1.11 | 16.49 | 1.03 | 11.21 | 0.99 |
| CEF, BIS, IBU | 22.21 | 0.85 | 15.78 | 0.98 | 8.09 | 0.99 | 4.23 | 1.09 |
| CEF, BIS, CALC | 80.26 | 0.58 | 24.36 | 0.74 | 13.11 | 0.72 | 5.45 | 1.23 |
| CEF, EST, CALC | 56.45 | 0.51 | 27.61 | 1.15 | 12.71 | 0.86 | 4.78 | 1.34 |

Where Xm—arithmetic mean, Sd—standard deviation.

After 2 h of observation, the content of antibiotic in any analyzed mixture did not exceed 14%. Only in a solution of cefepime alone and in combination with bisoprolol and calcium, after 30 min, was the amount of antibiotic still within 80%. A simultaneous blind test without the influence of radiation did not show any significant effect on changes in the stability of cefepime in tested drug combinations (Figure 4).

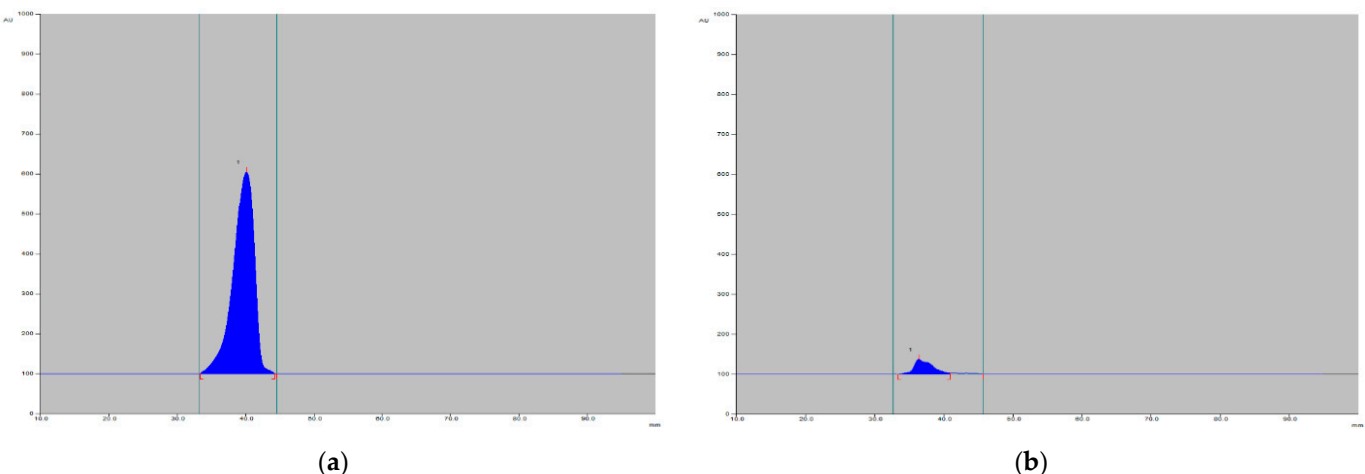

(**a**)                    (**b**)

**Figure 4.** Example densitograms registered for a model mixture with the composition 'CEF, EST, KET' after (**a**) 0 min, (**b**) 30 min, of UV irradiation.

Based on the obtained values, the order of reactions was determined. For this purpose, changes in the content of the cefepime during its incubation under the analyzed conditions were analyzed. Graphically illustrated data (in the form of plots of the dependence of ln[c] as a function of time, *t*), allowed us to show that changes in the cefepime concentration

over time occur with first-order kinetics (Figure 5), determined by the following differential equations [30]:

$$\frac{-d[S]}{dt} = k \times [S] \quad ln[S] = \ln [S]_0 - kt$$

where [S]—concentration [%], $[S]_0$—concentration [%] in time t = 0, $t$—time $[h]^{-1}$, $k$—the reaction rate constants $[h^{-1}]$.

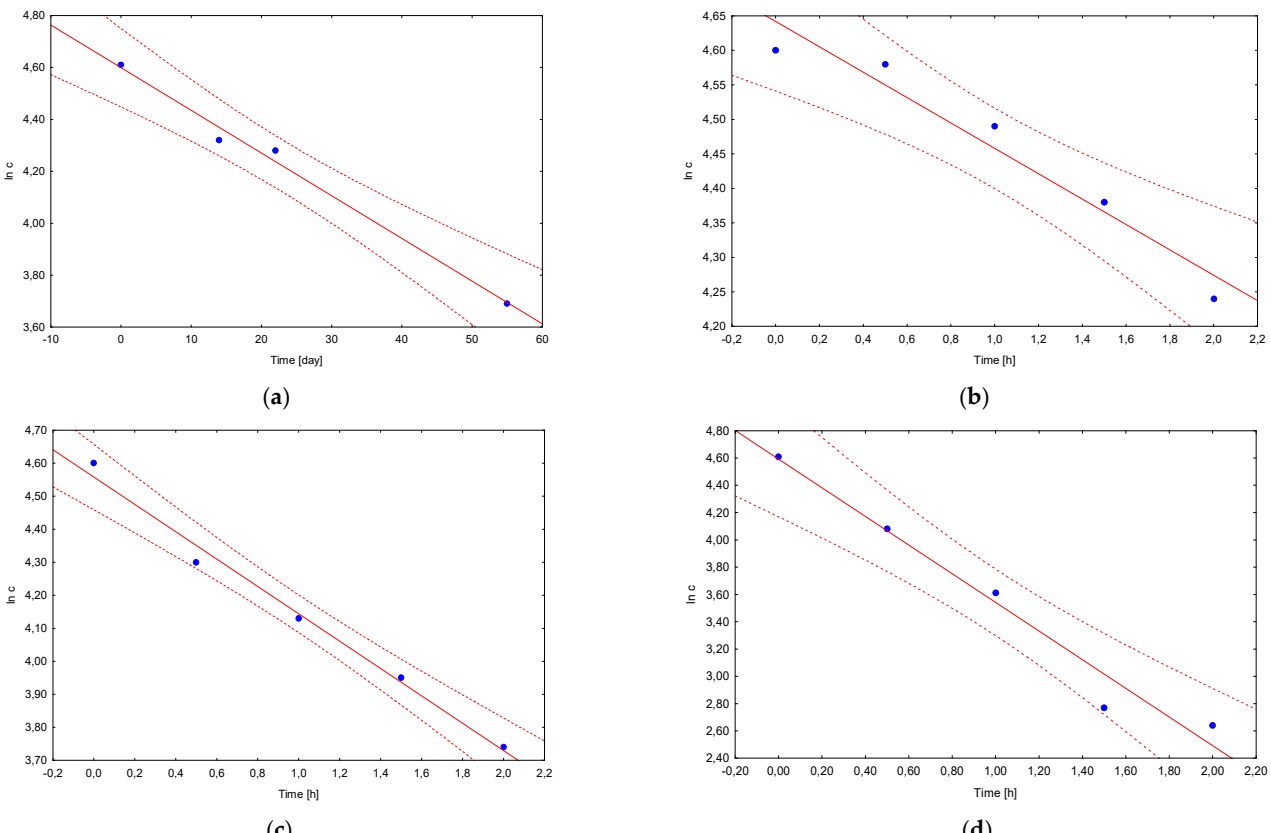

**Figure 5.** Selected graphs showing the degradation process of cefepime according to the first-order kinetics, (**a**) CEF at 4 °C, (**b**) CEF at 60 °C, (**c**) CEF, EST, and (**d**) CALC at 80 °C.

The reaction rate constants were calculated according to the equation:

$$k = \frac{-2.303 \, (logc1 - logc2)}{t_2 - t_1} \tag{1}$$

where $c_1$, $c_2$—concentration [%] after $t_1$, $t_2$ time [h].

Changes in the concentration [%] of cefepime in the test solutions over time are characterized by an almost complete correlation of the results, which is confirmed by the parameters of the regression curves, presented in Tables 4–6. The correlation coefficients take negative values, and thus there is an almost exact negative linear relationship between the examined features. The values of the correlation coefficients (r, $r^2$), which are a measure of the strength of the linear relationship between the variables, are close to the absolute value of unity (in the range of 0.7948–0.9861 under UV irradiation, 0.9304–0.9960 at 4 °C, 0.8062–0.9804 at 40 °C, 0.8271–0.9722 at 60 °C, and 0.8877–0.9950 at 80 °C), which, according to statistical nomenclature, means that the variables are almost completely correlated.

**Table 4.** Statistical parameters of calibration curves of cefepime in the tested drug mixtures at 4 °C.

| Mixture | a | b | Sa | Sb | Se | $R^2$ |
|---|---|---|---|---|---|---|
| CEF | −0.0164 | 4.5991 | 0.0012 | 0.0351 | 0.0466 | 0.9903 |
| CEF, KET | −0.0318 | 4.5563 | 0.0014 | 0.0436 | 0.0580 | 0.9960 |
| CEF, KET, CALC | −0.0197 | 4.6745 | 0.0025 | 0.0765 | 0.1017 | 0.9683 |
| CEF, GES + ETHYN | −0.0421 | 4.4320 | 0.0047 | 0.1439 | 0.1912 | 0.9753 |
| CEF, EST | −0.0380 | 4.5275 | 0.0029 | 0.0873 | 0.1159 | 0.9888 |
| CEF, EST, KET | −0.0572 | 4.4193 | 0.0062 | 0.1881 | 0.2499 | 0.9772 |
| CEF, GES + ETHY, KET | −0.0529 | 4.4042 | 0.0061 | 0.1847 | 0.2454 | 0.9744 |
| CEF, GES + ETHYN, CALC | −0.0236 | 4.6635 | 0.0034 | 0.1043 | 0.1385 | 0.9594 |
| CEF, BIS | −0.0278 | 4.5621 | 0.0017 | 0.0520 | 0.0691 | 0.9925 |
| CEF, COF | −0.0223 | 4.6133 | 0.0015 | 0.0459 | 0.0609 | 0.9910 |
| CEF, KOF, PAR | −0.0184 | 4.6009 | 0.0016 | 0.0484 | 0.0642 | 0.9853 |
| CEF, COF, ASA | −0.0477 | 4.5887 | 0.0027 | 0.0809 | 0.1075 | 0.9938 |
| CEF, ASA | −0.0395 | 4.5804 | 0.0022 | 0.0654 | 0.0869 | 0.9941 |
| CEF, BIS, PAR | −0.0193 | 4.5280 | 0.0023 | 0.0706 | 0.0938 | 0.9718 |
| CEF, BIS, IBU | −0.0470 | 4.4739 | 0.0036 | 0.1104 | 0.1466 | 0.9882 |
| CEF, BIS, CALC | −0.0136 | 4.7067 | 0.0026 | 0.0800 | 0.1063 | 0.9304 |
| CEF, EST, CALC | −0.0167 | 4.7321 | 0.0032 | 0.0989 | 0.1314 | 0.9295 |

The time needed to reduce the concentration of the reacting substance to half of the original value ($t_{0.5}$), and the time for the decomposition to reach 10% of the original value ($t_{0.1}$) were calculated according to the formulas [27]:

$$t_{0.1} = \frac{0.1053}{k} \qquad t_{0.5} = \frac{0.693}{k}$$

where: k—the reaction rate constants, $t_{0.1}$—the time during which the concentration decreases by 10%, $t_{0.5}$—the time during which the concentration decreases by 50%, and $C_1$, $C_2$—cefepime concentrations [%] after $t_1$ and $t_2$ time, $t_1$, $t_2$—time [h].

Changes in the concentration of cefepime over time, depending on the composition of the mixture tested, proceed at different rates. The reaction rate constants (k) increase with increasing temperature and assume higher values in 'CEF, GES + ETHN' and 'CEF, EST' solutions, while lower values occur in solutions where cefepime is stabilized by calcium ions (such as 'CEF, KET, CALC', 'CEF, BIS, CALC' and 'CEF, EST, CALC'), and in the presence of NSAIDs, paracetamol and caffeine ('CEF, KOF', 'CEF, KOF, PAR', 'CEF, KOF, ASA' and 'CEF, BIS, PAR'). The calculated values of $t_{0.5}$ and $t_{0.1}$ confirm greater stability of cefepime in solutions containing calcium ions, acetylsalicylic acid, and paracetamol, compared to antibiotic solutions with hormones and their combinations with ketoprofen.

Changes in the reaction rate constant (lnk) from the reciprocal of the temperature (1/T) occur similarly in all solutions, regardless of their composition. They differ only in the reaction-rate-constant values, which are higher in the solutions 'CEF, KET', 'CEF, GES + ETHN', 'CEF, EST', 'CEF, EST, KET', 'CEF, GES + ETYN, KET', 'CEF, BIS' and 'CEF, BIS, IBU'. The values of the rate constant at 40 °C are in the range of 0.09–0.36 h$^{-1}$, while in the case of exposure of drug mixtures to UV radiation they increase to 0.98–3.00 h$^{-1}$. The half-life for individual solutions treated with 80 °C is within the range of 0.30–2.39 h, for 60 °C it has higher values, in the range of 0.68–3.85 h, while lower-temperature conditions (4 °C) allowed this range to significantly extend, up to 284.02–1320 h, depending on the composition of the mixture. During the incubation of the samples at 80 °C, the $t_{0.1}$ values fluctuated in the range of 0.05–0.36 h, reaching values of 0.04–0.11 h for solutions under the influence of ultraviolet radiation (Table 7).

**Table 5.** Statistical parameters of calibration curves of cefepime in the tested drug mixtures at 40, 60 and 80 °C.

| Mixture | a | | | b | | | Sa | | | Sb | | | Se | | | $R^2$ | | |
|---|---|---|---|---|---|---|---|---|---|---|---|---|---|---|---|---|---|---|
| | 40 °C | 60 °C | 80 °C | 40 °C | 60 °C | 80 °C | 40 °C | 60 °C | 80 °C | 40 °C | 60 °C | 80 °C | 40 °C | 60 °C | 80 °C | 40 °C | 60 °C | 80 °C |
| CEF | −0.2320 | −0.2320 | −0.7880 | 4.6680 | 4.6680 | 4.7300 | 0.0450 | 0.0450 | 0.1618 | 0.0550 | 0.0551 | 0.1982 | 0.0711 | 0.0711 | 0.2558 | 0.8987 | 0.8987 | 0.8877 |
| CEF, KET | −0.2840 | −0.7080 | −0.9320 | 4.5720 | 4.7480 | 4.4140 | 0.0232 | 0.1304 | 0.1373 | 0.0284 | 0.1597 | 0.1681 | 0.0367 | 0.2061 | 0.2170 | 0.9804 | 0.9077 | 0.9389 |
| CEF, KET, CALC | −0.2080 | −0.3880 | −1.2060 | 4.6320 | 4.6120 | 4.7560 | 0.0218 | 0.0379 | 0.1705 | 0.0267 | 0.0464 | 0.2089 | 0.0344 | 0.0599 | 0.2696 | 0.9681 | 0.9722 | 0.9434 |
| CEF, GES + ETHYN | −0.2980 | −0.6260 | −1.8140 | 4.5580 | 4.6660 | 4.6460 | 0.0393 | 0.1165 | 0.1769 | 0.0481 | 0.1426 | 0.2166 | 0.0622 | 0.1841 | 0.2796 | 0.9503 | 0.9059 | 0.9723 |
| CEF, EST | −0.2860 | −1.0480 | −1.7560 | 4.5660 | 4.8380 | 4.7840 | 0.0656 | 0.2614 | 0.2927 | 0.0804 | 0.3202 | 0.3585 | 0.1038 | 0.4134 | 0.4628 | 0.8636 | 0.8427 | 0.9231 |
| CEF, EST, KET | −0.2420 | −1.0040 | −1.9960 | 4.5580 | 4.7120 | 4.2400 | 0.0385 | 0.1304 | 0.3338 | 0.0471 | 0.1587 | 0.4087 | 0.0609 | 0.2061 | 0.5277 | 0.9295 | 0.9519 | 0.9226 |
| CEF, GES + ETHYN, KET | −0.2420 | −0.9740 | −2.2600 | 4.5020 | 4.7121 | 4.4540 | 0.0684 | 0.1599 | 0.0922 | 0.0839 | 0.1959 | 0.1129 | 0.1083 | 0.2528 | 0.1457 | 0.8062 | 0.9252 | 0.9950 |
| CEF, GES + ETHYN, CALC | −0.2340 | −0.3140 | −0.7480 | 4.6020 | 4.600 | 4.5920 | 0.0556 | 0.0457 | 0.0330 | 0.0681 | 0.0559 | 0.0404 | 0.0879 | 0.0722 | 0.0522 | 0.8551 | 0.9404 | 0.9942 |
| CEF, BIS | −0.4100 | −0.7480 | −1.1820 | 4.5621 | 4.6680 | 4.5660 | 0.0974 | 0.0917 | 0.0902 | 0.1194 | 0.1123 | 0.1104 | 0.1541 | 0.1450 | 0.1425 | 0.8550 | 0.9569 | 0.9829 |
| CEF, COF | −0.0940 | −0.2240 | −0.5720 | 4.6133 | 4.5560 | 4.5460 | 0.0258 | 0.0433 | 0.0582 | 0.0316 | 0.0530 | 0.0712 | 0.0408 | 0.0684 | 0.0919 | 0.8157 | 0.8993 | 0.9699 |
| CEF, KOF, PAR | −0.0860 | −0.2840 | −0.5780 | 4.6009 | 4.5740 | 4.5740 | 0.0171 | 0.0416 | 0.0308 | 0.0210 | 0.0509 | 0.0377 | 0.0270 | 0.0657 | 0.0487 | 0.8941 | 0.9396 | 0.9916 |
| CEF, COF, ASA | −0.1300 | −0.4600 | −1.1480 | 4.5887 | 4.4920 | 4.5980 | 0.0290 | 0.0782 | 0.0514 | 0.0356 | 0.0958 | 0.0630 | 0.0460 | 0.1237 | 0.0813 | 0.8693 | 0.9202 | 0.9940 |
| CEF, ASA | −0.1300 | −0.3060 | −0.8100 | 4.5804 | 4.4960 | 4.5900 | 0.0351 | 0.0742 | 0.0650 | 0.0430 | 0.0909 | 0.0797 | 0.0555 | 0.1173 | 0.1029 | 0.8207 | 0.8500 | 0.9810 |
| CEF, BIS, PAR | −0.1160 | −0.1960 | −0.3920 | 4.5280 | 4.5220 | 4.5100 | 0.0174 | 0.0517 | 0.0704 | 0.0213 | 0.0634 | 0.0862 | 0.0276 | 0.0818 | 0.1112 | 0.9365 | 0.8271 | 0.0119 |
| CEF, BIS, IBU | −0.1400 | −0.4120 | −1.2420 | 4.4739 | 4.5020 | 4.4300 | 0.0271 | 0.0689 | 0.1569 | 0.0332 | 0.0844 | 0.1922 | 0.0429 | 0.1090 | 0.2481 | 0.8988 | 0.9225 | 0.9543 |
| CEF, BIS, CALC | −0.1260 | −0.1720 | −0.2780 | 4.7067 | 4.5660 | 4.6080 | 0.0232 | 0.0257 | 0.0154 | 0.0284 | 0.0315 | 0.0189 | 0.0367 | 0.0407 | 0.0244 | 0.9078 | 0.9372 | 0.9908 |
| CEF, EST, CALC | −0.1360 | −0.1700 | −0.4180 | 4.7321 | 4.5620 | 4.5640 | 0.0332 | 0.0279 | 0.0276 | 0.0408 | 0.0341 | 0.0338 | 0.0525 | 0.0441 | 0.0436 | 0.8481 | 0.9253 | 0.9935 |

**Table 6.** Statistical parameters of calibration curves of cefepime in the tested drug mixtures after UV radiation.

| Mixture | a | b | Sa | Sb | Se | $R^2$ |
|---|---|---|---|---|---|---|
| CEF | −1.2560 | 4.8260 | 0.1527 | 0.1870 | 0.2415 | 0.9575 |
| CEF, KET | −1.5140 | 4.3320 | 0.1753 | 0.2148 | 9.2772 | 0.9613 |
| CEF, KET, CALC | −1.4860 | 4.7240 | 0.1439 | 0.1763 | 0.2276 | 0.9726 |
| CEF, GES + ETHYN | −2.4000 | 4.8150 | 0.6465 | 0.6047 | 0.7228 | 0.8733 |
| CEF, EST | −1.8260 | 4.2770 | 0.4493 | 0.4203 | 0.5023 | 0.8920 |
| CEF, EST, KET | −3.0000 | 4.1700 | 1.5242 | 0.9839 | 1.0778 | 0.8048 |
| CEF, GES + ETHY, KET | −2.1780 | 4.6210 | 0.3823 | 0.3577 | 0.4275 | 0.9419 |
| CEF, GES + ETHYN, CALC | −2.0360 | 4.7870 | 0.4041 | 0.3780 | 0.4518 | 0.9270 |
| CEF, BIS | −1.4200 | 4.3860 | 0.1512 | 0.1852 | 0.2391 | 0.9671 |
| CEF, COF | −1.6440 | 4.5920 | 0.1956 | 0.2396 | 0.3093 | 0.9592 |
| CEF, KOF, PAR | −1.0500 | 4.5920 | 0.1083 | 0.1327 | 0.1713 | 0.9691 |
| CEF, COF, ASA | −1.5960 | 4.6460 | 0.1295 | 0.1586 | 0.2047 | 0.9806 |
| CEF, ASA | −2.4500 | 4.5000 | 0.3373 | 0.3155 | 0.3771 | 0.9635 |
| CEF, BIS, PAR | −1.0820 | 4.4180 | 0.1216 | 0.1489 | 0.1923 | 0.9636 |
| CEF, BIS, IBU | −1.4900 | 4.2660 | 0.2071 | 0.2536 | 0.3274 | 0.9452 |
| CEF, BIS, CALC | −1.5640 | 4.8320 | 0.1550 | 0.1899 | 0.2451 | 0.9714 |
| CEF, EST, CALC | −1.5980 | 4.7600 | 0.1097 | 0.1344 | 0.1735 | 0.9861 |

**Table 7.** Kinetics parameters designated for degradation processes of cefepime in various drug mixtures, under tested conditions.

| Mixture | UV | 4 °C | 40 °C | 60 °C | 80 °C |
|---|---|---|---|---|---|
| CEF | k = 1.26 $t_{0.1}$ = 0.08 $t_{0.5}$ = 0.55 | k = 6.94 × 10$^{-4}$ $t_{0.1}$ = 151.73 $t_{0.5}$ = 998.56 | k = 0.24 $t_{0.1}$ = 0.44 $t_{0.5}$ = 2.89 | k = 0.60 $t_{0.1}$ = 0.18 $t_{0.5}$ = 1.16 | k = 0.86 $t_{0.1}$ = 0.12 $t_{0.5}$ = 0.81 |
| CEF, KET | k = 1.61 $t_{0.1}$ = 0.07 $t_{0.5}$ = 0.43 | k = 1.34 × 10$^{-3}$ $t_{0.1}$ = 78.58 $t_{0.5}$ = 517.16 | k = 0.29 $t_{0.1}$ = 0.36 $t_{0.5}$ = 2.39 | k = 0.73 $t_{0.1}$ = 0.14 $t_{0.5}$ = 0.94 | k = 0.98 $t_{0.1}$ = 0.11 $t_{0.5}$ = 0.71 |
| CEF, KET, CALC | k = 1.50 $t_{0.1}$ = 0.07 $t_{0.5}$ = 0.46 | k = 7.74 × 10$^{-4}$ $t_{0.1}$ = 136.05 $t_{0.5}$ = 895.35 | k = 0.19 $t_{0.1}$ = 0.55 $t_{0.5}$ = 3.65 | k = 0.37 $t_{0.1}$ = 0.28 $t_{0.5}$ = 1.87 | k = 1.10 $t_{0.1}$ = 0.10 $t_{0.5}$ = 0.63 |
| CEF, GES + ETHYN | k = 2.61 $t_{0.1}$ = 0.04 $t_{0.5}$ = 0.27 | k = 1.82 × 10$^{-3}$ $t_{0.1}$ = 57.86 $t_{0.5}$ = 380.77 | k = 0.32 $t_{0.1}$ = 0.33 $t_{0.5}$ = 2.17 | k = 0.69 $t_{0.1}$ = 0.15 $t_{0.5}$ = 1.00 | k = 1.75 $t_{0.1}$ = 0.06 $t_{0.5}$ = 0.40 |
| CEF, EST | k = 2.00 $t_{0.1}$ = 0.05 $t_{0.5}$ = 0.35 | k = 1.61 × 10$^{-3}$ $t_{0.1}$ = 65.40 $t_{0.5}$ = 430.43 | k = 0.34 $t_{0.1}$ = 0.31 $t_{0.5}$ = 2.00 | k = 0.80 $t_{0.1}$ = 0.13 $t_{0.5}$ = 0.87 | k = 1.61 $t_{0.1}$ = 0.07 $t_{0.5}$ = 0.43 |
| CEF, EST, KET | k = 3.00 $t_{0.1}$ = 0.04 $t_{0.5}$ = 0.23 | k = 2.44 × 10$^{-3}$ $t_{0.1}$ = 43.16 $t_{0.5}$ = 284.02 | k = 0.24 $t_{0.1}$ = 0.44 $t_{0.5}$ = 2.89 | k = 1.02 $t_{0.1}$ = 0.10 $t_{0.5}$ = 0.68 | k = 1.96 $t_{0.1}$ = 0.05 $t_{0.5}$ = 0.36 |
| CEF, GES + ETHYN, KET | k = 2.34 $t_{0.1}$ = 0.05 $t_{0.5}$ = 0.30 | k = 2.27 × 10$^{-3}$ $t_{0.1}$ = 46.39 $t_{0.5}$ = 305.29 | k = 0.29 $t_{0.1}$ = 0.36 $t_{0.5}$ = 2.39 | k = 1.02 $t_{0.1}$ = 0.10 $t_{0.5}$ = 0.68 | k = 2.30 $t_{0.1}$ = 0.05 $t_{0.5}$ = 0.30 |
| CEF, GES + ETHYN, CALC | k = 2.15 $t_{0.1}$ = 0.05 $t_{0.5}$ = 0.32 | k = 9.38 × 10$^{-4}$ $t_{0.1}$ = 112.26 $t_{0.5}$ = 738.81 | k = 0.21 $t_{0.1}$ = 0.50 $t_{0.5}$ = 3.30 | k = 0.29 $t_{0.1}$ = 0.36 $t_{0.5}$ = 2.39 | k = 0.73 $t_{0.1}$ = 0.14 $t_{0.5}$ = 0.95 |
| CEF, BIS | k = 1.50 $t_{0.1}$ = 0.07 $t_{0.5}$ = 0.50 | k = 1.18 × 10$^{-3}$ $t_{0.1}$ = 89.24 $t_{0.5}$ = 587.29 | k = 0.36 $t_{0.1}$ = 0.29 $t_{0.5}$ = 1.93 | k = 0.78 $t_{0.1}$ = 0.14 $t_{0.5}$ = 0.89 | k = 1.15 $t_{0.1}$ = 0.09 $t_{0.5}$ = 0.60 |
| CEF, COF | k = 1.75 $t_{0.1}$ = 0.06 $t_{0.5}$ = 0.40 | k = 9.38 × 10$^{-4}$ $t_{0.1}$ = 112.26 $t_{0.5}$ = 738.81 | k = 0.09 $t_{0.1}$ = 1.17 $t_{0.5}$ = 7.70 | k = 0.26 $t_{0.1}$ = 0.41 $t_{0.5}$ = 2.67 | k = 0.60 $t_{0.1}$ = 0.18 $t_{0.5}$ = 1.16 |

**Table 7.** *Cont.*

| Mixture | UV | 4 °C | 40 °C | 60 °C | 80 °C |
|---|---|---|---|---|---|
| CEF, KOF, PAR | k = 0.98<br>$t_{0.1} = 0.11$<br>$t_{0.5} = 0.71$ | k = 7.74 × 10⁻⁴<br>$t_{0.1} = 136.05$<br>$t_{0.5} = 895.35$ | k = 0.09<br>$t_{0.1} = 1.17$<br>$t_{0.5} = 7.70$ | k = 0.32<br>$t_{0.1} = 0.33$<br>$t_{0.5} = 2.17$ | k = 0.59<br>$t_{0.1} = 0.18$<br>$t_{0.5} = 1.17$ |
| CEF, COF, ASA | k = 1.50<br>$t_{0.1} = 0.07$<br>$t_{0.5} = 0.50$ | k = 2.01 × 10⁻³<br>$t_{0.1} = 52.39$<br>$t_{0.5} = 344.78$ | k = 0.13<br>$t_{0.1} = 0.81$<br>$t_{0.5} = 5.33$ | k = 0.51<br>$t_{0.1} = 0.21$<br>$t_{0.5} = 1.36$ | k = 1.15<br>$t_{0.1} = 0.09$<br>$t_{0.5} = 0.60$ |
| CEF, ASA | k = 2.34<br>$t_{0.1} = 0.05$<br>$t_{0.5} = 0.30$ | k = 1.67 × 10⁻³<br>$t_{0.1} = 63.05$<br>$t_{0.5} = 414.97$ | k = 0.14<br>$t_{0.1} = 0.75$<br>$t_{0.5} = 4.95$ | k = 0.36<br>$t_{0.1} = 0.29$<br>$t_{0.5} = 1.93$ | k = 0.86<br>$t_{0.1} = 0.12$<br>$t_{0.5} = 0.81$ |
| CEF, BIS, PAR | k = 1.10<br>$t_{0.1} = 0.10$<br>$t_{0.5} = 0.63$ | k = 8.40 × 10⁻⁴<br>$t_{0.1} = 125.36$<br>$t_{0.5} = 825.00$ | k = 0.12<br>$t_{0.1} = 0.88$<br>$t_{0.5} = 5.78$ | k = 0.22<br>$t_{0.1} = 0.48$<br>$t_{0.5} = 3.15$ | k = 0.39<br>$t_{0.1} = 0.27$<br>$t_{0.5} = 1.78$ |
| CEF, BIS, IBU | k = 1.61<br>$t_{0.1} = 0.07$<br>$t_{0.5} = 0.50$ | k = 2.01 × 10⁻³<br>$t_{0.1} = 52.39$<br>$t_{0.5} = 344.78$ | k = 0.14<br>$t_{0.1} = 0.75$<br>$t_{0.5} = 4.95$ | k = 0.46<br>$t_{0.1} = 0.23$<br>$t_{0.5} = 1.51$ | k = 1.20<br>$t_{0.1} = 0.09$<br>$t_{0.5} = 0.58$ |
| CEF, BIS, CALC | k = 1.50<br>$t_{0.1} = 0.07$<br>$t_{0.5} = 0.50$ | k = 5.25 × 10⁻⁴<br>$t_{0.1} = 200.57$<br>$t_{0.5} = 1320.00$ | k = 0.13<br>$t_{0.1} = 0.81$<br>$t_{0.5} = 5.33$ | k = 0.18<br>$t_{0.1} = 0.59$<br>$t_{0.5} = 3.85$ | k = 0.29<br>$t_{0.1} = 0.36$<br>$t_{0.5} = 2.39$ |
| CEF, EST, CALC | k = 1.61<br>$t_{0.1} = 0.07$<br>$t_{0.5} = 0.50$ | k = 6.39 × 10⁻⁴<br>$t_{0.1} = 164.79$<br>$t_{0.5} = 1084.51$ | k = 0.13<br>$t_{0.1} = 0.81$<br>$t_{0.5} = 5.33$ | k = 0.18<br>$t_{0.1} = 0.59$<br>$t_{0.5} = 3.85$ | k = 0.43<br>$t_{0.1} = 0.24$<br>$t_{0.5} = 1.61$ |

Analyzing the impact of various stress conditions in the form of UV radiation and the increase or decrease in the storage temperature of model cefepime-hydrochloride solutions allowed us to conclude that the content of cefepime in individual drug mixtures increases with a decrease in the temperature at which they are stored. The antibiotic showed the highest stability in connection with calcium, while the addition of gestodene, estradiol and their combinations with ketoprofen had the most destructive effect. Observations graphically illustrated, in the form of the relationship lnc = f(*t*) for individual mixtures, prove that changes in the rate of cefepime decomposition occur according to first-order kinetics, regardless of the composition and the conditions to which they are subjected.

Based on the obtained results, we can conclude that the varying rate of degradation of an antibiotic may be the result of the properties of another active substance or other excipient, carrier or vehicle which is incompatible with a component of the mixture.

## 4. Conclusions

The quantitative conditions for the determination of cefepime hydrochloride in solution individually and in mixtures containing the variable ketoprofen, gestodene with ethinylestradiol, estradiol, caffeine, calcium ions, paracetamol, bisoprolol, acetylsalicylic acid and ibuprofen, were established using the TLC technique with densitometric detection.

The data obtained during the experiments on cefepime stability in mixtures of different composition permit the conclusion that the stability of the antibiotic decreases with increasing temperature. The statistical investigation proved that the rate of change in cefepime concentration in all tested combinations and miscellaneous conditions (high and low temperature, incubation time, UV radiation) is consistent with the first-order kinetics. It was found that the highest stability of cefepime was recorded at 4 °C. The highest stability showed cefepime in combinations with calcium ions, whilst the quickest degradation of antibiotics occurs in connection with hormones (gestodene, estradiol) and their combinations with ketoprofen. Analogous relationships were observed for solutions subjected to UV radiation.

**Author Contributions:** Conceptualization, M.D. and J.Ż.; methodology, M.D. and J.Ż.; software, J.Ż.; formal analysis, Ż.B.-K. and M.S.; investigation, J.Ż. and Ż.B.-K.; writing—original draft preparation, J.Ż.; writing—review and editing, M.D. and M.S.; supervision, M.D. All authors have read and agreed to the published version of the manuscript.

**Funding:** This research received no external funding.

**Institutional Review Board Statement:** Not applicable.

**Data Availability Statement:** Not applicable.

**Conflicts of Interest:** The authors declare no conflict of interest.

**Abbreviations**

| | |
|---|---|
| ASA | acetylsalicylic acid |
| BIS | bisoprolol |
| CALC | calcium |
| CEF | cefepime |
| DDD | defined daily doses |
| EST | estradiol |
| GES + ETHYN | gestodene and ethinylestradiol |
| HRT | hormone-replacement therapy |
| IBU | ibuprofen |
| KET | ketonal |
| KOF | caffeine |
| NSAIDs | non-steroidal anti-inflammatory drugs |
| OTC | over-the-counter |
| PAR | paracetamol |
| PBP | penicillin-binding proteins |
| WHO | World Health Organization |

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
