# Peer review of "The Stability Study of Cefepime Hydrochloride in Various Drug Combinations"

_processes, doi:10.3390/pr11020602_

Round 1
Reviewer 1 Report
I have gone through the manuscript title "Cephalosporin’s therapy in the context of drug interactions".
Manuscript is well written. Need to concise abstract and add some results in the conclusion section.
add some graphical figures which shows some deviations in the readings mentioned in the table 1,2 and so on........ Also add deviations in the table too. (std dev.)
Write equations using equation tab in the word file. Number or words overlapping each other.
Author used maximum of abbreviations in the manuscript. Kindly add one abbreviation section before reference which will help the readers to understand.
Technically this manuscript shows an excellent impact on the study performed by the authors.
Author Response
-

Reviewer 2 Report
Joanna et al., did excellent investigations on cefepime that has potential for understanding stability-drug-drug interactions in a solution. Here you can find my comments to improve the quality and understanding of the works-
Major comments:
1. Author need to avoid “large paragraph” in the manuscript. There are numerous places where reference was missing and very broad meaningful statement that causing a “complexity” to the readers.
2. Author missed to show a solid background on which basis the hypothesis was generated (i.e. Line 217-219). What do you mean by maintenance of medications in combination? These medications usually never stored in a formulation. Please show some references.
3. Author did not included any clinical drug-drug interactions mechanism such as absorption, metabolism, transport, that usually refers to the ultimate outcome of the DDIs. Author should describe them to prepare a hypothesis and conclusion of the study.
4. The key outcome of this study needs to be verified and cross-validation with in vivo study. For example, at low temperature (line 371-372) Cefepime had 25% less stability. This is always should be known and labeled by the manufacturer. To the DDI perspective, this is very confusing to interpret and no relevance! We can not think after administration this antibiotic will face this condition in the gut, or GIT. Thus, application of this findings to the clinic is very misleading. Please author need to re-phrasing or deliver key message in regards to the outcomes.
5. Misleading statement on line 374-378, why this very long stability with DDI functions? The half-life of cefepime was approximately 2.3 hours in subjects with normal kidney function. Author needs to show any clinical use where this prolong period of exposure of all combination (‘CEF, EST, KET). Author needs to design and find the outcomes regarding physiological and clinical considerations. Otherwise, this study looks like a stability study not DDI focused!
Minor comments:
1. Please add correct references from where you cited. i.e. Line 35-36, 42-47, 50-63, 211-216.
Author Response
-

Round 2
Reviewer 2 Report
Thank you for addressing review comments and revising manuscript title in relevant to the main outcomes of this study.
Author Response
-
